# Biochemical and Functional Profiling of Thioredoxin-Dependent Cytosolic GPX-like Proteins in *Euglena gracilis*

**DOI:** 10.3390/biom14070765

**Published:** 2024-06-27

**Authors:** Md Topu Raihan, Takahiro Ishikawa

**Affiliations:** 1The United Graduate School of Agricultural Sciences, Tottori University, 4-101 Koyama-Minami, Tottori 680-8550, Japan; topuraihan09sust@gmail.com; 2Institute of Agricultural and Life Sciences, Academic Assembly, Shimane University, 1060 Nishikawatsu, Matsue 690-8504, Japan

**Keywords:** *Euglena gracilis*, cytosolic GPX-like protein, reactive oxygen species

## Abstract

Unlike plants and animals, the phytoflagellate *Euglena gracilis* lacks catalase and contains a non-selenocysteine glutathione peroxidase-like protein (EgGPXL), two peroxiredoxins (EgPrx1 and EgPrx4), and one ascorbate peroxidase in the cytosol to maintain reactive oxygen species (ROS) homeostasis. In the present study, the full-length cDNA of three cytosolic EgGPXLs was obtained and further characterized biochemically and functionally. These EgGPXLs used thioredoxin instead of glutathione as an electron donor to reduce the levels of H_2_O_2_ and *t*-BOOH. The specific peroxidase activities of these enzymes for H_2_O_2_ and *t*-BOOH were 1.3 to 4.9 and 0.79 to 3.5 µmol/min/mg protein, respectively. Cytosolic *EgGPXL*s and *EgPrx1*/*EgPrx4* were silenced simultaneously to investigate the synergistic effects of these genes on the physiological function of *E. gracilis*. The suppression of cytosolic *EgGPXL* genes was unable to induce any critical phenomena in *Euglena* under normal (100 μmol photons m^−2^ s^−1^) and high-light conditions (350 μmol photons m^−2^ s^−1^) at both autotrophic and heterotrophic states. Unexpectedly, the suppression of *EgGPXL* genes was able to rescue the *EgPrx1/EgPrx4*-silenced cell line from a critical situation. This study explored the potential resilience of *Euglena* to ROS, even with restriction of the cytosolic antioxidant system, indicating the involvement of some compensatory mechanisms.

## 1. Introduction

Antioxidant systems are one of the most sophisticated and crucial cellular systems that enable every organism to withstand unfavorable conditions during different types of biotic and abiotic stresses [1,2]. Usually, the antioxidant system is activated to maintain redox homeostasis (redoxtasis) when reactive oxygen species (ROS) levels increase to toxic levels. Nevertheless, ROS acts as secondary messenger molecules in response to growth and development, as well as during biotic and abiotic stresses in an organism [3]. ROS-like superoxide radicals (O_2_^−^), hydrogen peroxide (H_2_O_2_), and singlet oxygen (^1^O_2_), produced in response to different types of environmental stimuli, can cause severe damage to biomolecules including DNA, proteins, and lipids under conditions of elevated ROS levels [1]. The antioxidant system is activated and controlled by various enzymatic and non-enzymatic biomolecules. Enzymatic biomolecules, such as catalase, peroxidase, glutaredoxin, thioredoxin (Trx), ascorbate peroxidase (APX), tryparedoxin peroxidase, and superoxide dismutase, have been reported in various prokaryotic and eukaryotic organisms [4,5]. These enzymatic molecules are strategically localized across different cellular compartments and neutralize ROS, which may be detrimental to macromolecules and lead to deregulated cellular metabolism and physiological function [1,4]. Specifically, ROS are produced in photosynthetic organisms owing to CO_2_ deficiency, a phenomenon that leads to competitive reactions. In this reaction, O_2_ becomes the favored substrate over CO_2_, which ultimately governs oxygenation and the subsequent production of ROS [1]. The produced ROS can be detected in photosynthetic organisms through at least three mechanisms, i.e., some mysterious or unknown protein receptors, ROS-dependent inhibition of phosphatases, and some redox-sensitive transcription factors, namely NPR1 (non-expressor of pathogenesis-related proteins 1) or HSFs (heat shock factors) [6].

Glutathione peroxidase (GPX) and peroxiredoxin (Prx), collectively known as thiol/selenol peroxidases or non-heme peroxidases, are regarded as key players in the antioxidant system because of their emerging efficiency in eliminating cellular hydrogen peroxide (H_2_O_2_), organic hydroperoxides (*t*-BOOH), and lipid peroxides to maintain the thiol state across the cellular environment [7,8]. Unlike mammals, the former thiol peroxidase derived from some photosynthesizing organisms, as well as other domains of life, use Trx instead of reduced tripeptide thiol glutathione (γ-glutamyl-cysteinyl-glycine, GSH) as an electron donor, which may be occurred due the substitution of the active amino acid selenocysteine to cysteine in these organisms [9,10]. Selenocysteine is a rare amino acid encoded by the opal codon UGA and is usually found in certain domains of organisms [11,12]. Interestingly, the opal codon UGA is not recognized as a stop codon in these organisms due to the presence of a cis-acting element called the selenocysteine insertion sequence (SECIS) [12]. Consequently, the selenocysteine-deficient GPX group is regarded as non-selenium GPX (NS-GPX), which might be involved in cellular cross-communication during abiotic and biotic stresses through direct protein–protein interactions, redox-mediated signal transduction, transcription factors, and epigenetic control [10]. Most plant GPXs are monomeric, excluding poplar GPX-5, which has been reported as a dimer owing to its unique hydrophobic and aromatic networks in the redox state [13]. Plant GPX usually lacks amino acids to encode the oligomerization loop found in classical GPX, whereas this loop is found in all mammalian GPX except GPX-4 [14]. Therefore, GPX-4 is the only mammalian GPX found as a monomer, which can also reduce intracellular phospholipid hydroperoxides directly, unlike other GPX proteins [15]. In contrast to plant GPX, the photosynthetic green algae *Chlamydomonas reinhardtii* possesses selenium-containing GPX (CrGPX-1 and CrGPX-2) along with NS-GPX (CrGPX-3, -4, and -5). The SECIS elements, which are responsible for the presence of selenium-containing GPX in this organism, are similar to those in mammals but not identical [11]. Formerly, GPX proteins were regarded as ROS scavengers, but recently, their roles in various cellular functions have been reported, as mentioned above. For example, the interaction of AtGPX-3 with the 2C-type protein phosphatase abscisic acid insensitive 1 and 2 (ABI1 and 2) induces the activation of plasma membrane Ca^2+^ and K^+^ channels, leading to stomatal closure during ABA and drought stress responses [16]. In addition, NS-GPX interacts with dehydration-responsive element-binding proteins (DREB2A and DREB2B) through another transcriptional regulator protein, CEO1, to regulate genes involved in dehydration and heat stress in *Arabidopsis* [16]. Furthermore, AtGPX-8, localized in the nucleus and cytosol, interacts with redox modification proteins through retrograde signaling, which further assists the transport of AtGPX-8 to the nucleus to protect nuclear components [17]. Transgenic tobacco plants are resistant to osmotic, oxidative, and chilling stresses when *CrGPX-5* is overexpressed [11]. Accordingly, *Arabidopsis* acquires resilience to abiotic stress after overexpressing cyanobacterium *Synechocystis* GPX-like proteins [18]. Interestingly, proteomic analysis reveals that the suppression of *OsGPX-3* negatively manipulates histone acetylase enzymes and enzymes involved in DNA processing, indicating its association with histone modification [19]. 

The photosynthesizing, flagellated, single-celled, free-living, eukaryotic protist *Euglena gracilis* harbors the key characteristics of both autotrophic and heterotrophic organisms, making it one of the most interesting organisms in the scientific world. Evolutionarily, it is considered to be the supergroup *Discoba* under the phylum *Euglenazoa*, which also includes some important parasitic organisms, such as *Trypanosoma brucei* and *Leishmania donovani* [20]. *E. gracilis* possesses a very rare antioxidant system to maintain redox homeostasis across the cellular level compared to plants and animals. It is deficient in catalase and harbors a trypanothione system that is uncommon in other domains of life [21]. Additionally, it contains water-soluble antioxidants L-ascorbic acid (AsA), GSH, hydrophobic carotenoids, and tocopherols [22]. Furthermore, a single cytosol-centric ascorbate peroxidase (APX) has been identified throughout the genome of *E. gracilis*, which forms an intermolecular dimeric structure through two homologous catalytic domains [23]. It can efficiently reduce both H_2_O_2_ and *t*-BOOH, and silencing the *APX* gene increases the amount of H_2_O_2_ in *E. gracilis* cells, indicating its potential role in ROS metabolism [23]. In addition, two cytosolic (EgPrx-1, -4), one mitochondrial (EgPrx-3), and one chloroplastic (EgPrx-2) peroxiredoxins have been characterized in *E. gracilis*, and the silencing of *EgPrx-1, -4* induces critical phenomena under both heterotrophic and autotrophic condition [24]. Similarly, silencing the cytosolic NADPH-dependent thioredoxin reductase *EgNTR-2* inhibits the growth of *E. gracilis* significantly [25]. In addition, glutathione reductase (GR) and trypanothione reductase (TR) are also identified without any functional analysis. Notably, TR is found in trypanosomatids instead of GR, but interestingly, *Euglena* possesses both types of reductases, suggesting the presence of a complex antioxidant system [21]. 

Although transcriptomic analysis has revealed the involvement of numerous enzymes in *Euglena*’s antioxidant system, only a few enzymes have been characterized biochemically and functionally, indicating a lack of information regarding the antioxidant enzymes of this organism, specifically non-selenocysteine GPX-like proteins (EgGPXLs) [22,26]. Therefore, we focused on characterizing EgGPXLs using biochemical and functional analyses. According to our previous transcriptomic data, *E. gracilis* encodes four types of GPXLs, designated EgGPXL-1~EgGPXL-4 [26]. Of these, only chloroplastic EgGPXL-1 has been characterized biochemically and functionally, and the silencing of this chloroplastic GPXL was unable to trigger any phenotypic change under stress conditions in both heterotrophic and autotrophic states in *Euglena* [27]. The remaining EgGPXL-2, -3, and -4 are anticipated to be localized within the cytosol, where EgPrx-1 and EgPrx-4 are also located and proved to be critical under both heterotrophic and autotrophic conditions [24]. Eventually, the biochemical and functional characterization of these cytosolic EgGPXLs, as well as the simultaneous silencing of *EgPrx-1* and *EgPrx-4* genes with these *EgGPXL*s, will shed light on the regulation of cytosol-centric Trx-dependent peroxidases in *Euglena* involved in redoxtasis.

## 2. Materials and Methods

### 2.1. Cell Culture

*E. gracilis* strain Z was maintained by routine subculturing in heterotrophic Koren–Hutner (KH) medium (pH-3.5). In order to obtain optimum number of cells, the cells were incubated at 26 °C under 120 rpm with continuous illumination of 50 µmol photons m^−2^ s^−1^ in a 50 mL Erlenmeyer flask for six days. Similar incubation conditions were used for autotrophic growth using Cramer–Myers (CM) medium, but the cells were incubated for two weeks to reach the stationary phase. When the cells reached the stationary phase, they were transferred to fresh medium. Cells were counted using an electric field multichannel cell counting technology called CASY (Roche Diagnostics, Basel, Switzerland).

### 2.2. RNA Extraction from Euglena 

Total RNA was extracted using RNAiso Plus (Takara, Shiga, Japan), following the manufacturer’s instructions. Briefly, 1 mL of 5-day-old *Euglena* culture was centrifuged at 20,000× *g* for 2 min. The cells were vortexed for 5 min after the addition of 1 mL of RNAiso Plus. After that, the cells were kept standstill for 5 min, and 200 µL chloroform was added. The cells were mixed with chloroform, vortexed for 1 min, and allowed to stand for 1 min. The solution was then centrifuged at 20,000× *g* for 15 min at room temperature. Then, 400 µL of supernatant was transferred to 1.5 mL eppendorf tube, and 1 mL of isopropanol was added. Afterward, the sample was mixed gently and further centrifuged at 20,000× *g* for 15 min at room temperature after allowing it to stand for 15 min. The supernatant was removed and the pellet was washed carefully with 70% ethanol. Finally, 50 µL of TE (Tris-EDTA) buffer was added to the pellet after completely removing the ethanol. The RNA quality was checked using a Trace UV–visible spectrophotometer (Tomy Seiko/Q5000, Tokyo, Japan).

### 2.3. Cloning of Cytosolic EgGPXLs

After obtaining RNA from *Euglena*, it was reverse transcribed into cDNA using PrimeScript™ Reverse Transcription reagent Kit according to manufacturer instructions (Takara, Shiga, Japan). Full-length cDNA encoding mature EgGPXL-2, -3, and -4 were obtained by PCR, using primers attached along with infusion site (Appendix A). The target amplicon and pCold-II vector (Takara) were infused through a process called infusion cloning using In-Fusion HD cloning kit (Takara) to produce 6X-His-tagged recombinant proteins. Subsequently, the assumed recombinant vector was transferred to *E. coli* DH5α for the confirmation of cloning. Transformation was conducted with the addition of 1 µL infusion mixture in 200 µL *E. coli* DH5α competent cell, and next, the mixture was mixed gently. Then, the solution was kept on ice for 10 min and subsequently taken for heat shock at 42 °C. Afterward, it was kept on ice for 2 min, and then 400 µL of SOC media was added to the mixture. After that, it was incubated for 1 h at 37°C. After 1 h incubation, the liquid culture of the cells was spread on LB agar (Luria-Bertani agar: 1% polypeptone, 0.5% yeast extract, and 0.8% NaCl, 1.5% agar; pH-7) plate supplemented with 50 μg ml^−1^ ampicillin and incubated for overnight at 37 °C. Finally, the desired colonies were incubated overnight in LB broth supplemented with the same antibiotic. Eventually, plasmid extraction was conducted from the overnight culture according to manufacturer instructions using HiYield™ Plasmid Mini Kit (RBC Bioscience, New Taipei, Taiwan) from desired colony. After that presence of the complete ORF of target genes in recombinant pCold-II vector was confirmed by Sanger sequencing on an automatic DNA sequencer (ABI PRISMTM 3130xl, Applied Biosystems, Waltham, MA, USA). For protein expression analysis, each recombinant vector was transferred into *E. coli* strain BL21 Star (Agilent Technologies, Santa Clara, CA, USA) using the protocol mentioned above.

### 2.4. Expression and Purification of Recombinant Proteins

After the successful transformation of the target recombinant vector in *E. coli* BL21, the desired colony was incubated in LB medium at 37 °C for 16 h. The medium was supplemented with 50 μg mL^−1^ ampicillin and 34 μg mL^−1^ of chloramphenicol to grow *E. coli* BL21 within desired recombinant vector. IPTG (isopropyl β-D-thiogalactopyranoside) was subsequently supplemented at concentration of 0.5 mM to induce the expression of target protein after achieving the absorbance of 0.4–0.5 at 600 nm of the cultured cells. Afterward, the culture was maintained at 4 °C for 30 min. The cells were further taken in incubation for large-scale recombinant EgGPXL-2, -3, -4 production at 15 °C for 20 h. Following completion of overnight incubation of culture, it was collected through centrifugation at 20,000× *g* for 10 min (4 °C). The collected cells were first resuspended in 50 mM Tris-HCl buffer (pH 7.5) and disrupted by sonication for 5 min (10 kHz, using 30 s strokes with 30 s intervals) for purification. Consequently, the lysed cells were taken in centrifugation to obtain the recombinant protein lysate using 20,000× *g*, 15 min (4 °C) centrifugation condition. TALON Metal Affinity resin (Clontech, Palo Alto, CA, USA) was used for purification of target recombinant Hexa-His-tagged protein and subsequently stored at 4 °C for further use. Refrigeration temperature was strictly maintained during protein purification. Each fraction obtained from protein purification was subjected to 15% SDS-PAGE to check the expression and presence of the purified recombinant protein. The purified protein was quantified by Bradford protein assay [28].

### 2.5. Detection of Recombinant EgGPXLs (rEgGPXLs) Activity

The peroxidase activity of purified recombinant protein was determined by measuring NADPH oxidation in absorbance at 340 nm (ε = 6.22 mM^−1^ cm^−1^) according to previously described protocol [29]. The reaction mixture was prepared using 0.2 mM NADPH, 6 μM Trx (Recombinant yeast Trx-2, Oriental Yeast, Tokyo, Japan), and 0.3 μM recombinant yeast Trx reductase (Oriental Yeast, Tokyo, Japan) in 20 mM HEPES-NaOH, pH 7.5, in a final volume of 0.1 mL. Two different electron acceptors, H_2_O_2_ and *t*-BOOH, were utilized up to 0.04 mM to investigate the peroxidase activity. 

### 2.6. Preparation of Crude Extract of Euglena

*E. gracilis* cells were collected at optimal growth and centrifuged at 20,000× *g* for 10 min (4 °C). The cells were resuspended in 50 mM ice-cold Tris-HCl buffer (pH 7.5) and sonicated for disruption of the cells (10 kHz, 30 s strokes with 30 s intervals). After centrifugation under the same conditions, the supernatant was used for APX and peroxidase activity assays. APX activity was measured as a decrease in absorbance at 290 nm (ε = 2.80 mM^−1^·cm^−1^), as described previously [23]. The reaction mixture for APX activity consisted of 50 mM potassium phosphate buffer (pH 7.0), 1 mM EDTA, 0.4 mM AsA, and 0.1 mM H₂O₂. Peroxidase activity of the *Euglena* lysate was determined as previously described [24].

### 2.7. Generation of Cytosolic EgGPXL Mutants through RNA Interference 

*EgGPXL* suppression was performed using RNA interference (RNAi), as described previously [30]. First, a partial cDNA sequence of approximately 500 bp was amplified by PCR using a specific primer set containing an additional T7 RNA polymerase promoter sequence at 5’-ends (Appendix A). Furthermore, sense and antisense RNAs were obtained from the amplified PCR products following the manufacturer’s instructions using the MEGAscript RNAi Kit (Life Technologies, Carlsbad, CA, USA). Transcribed RNA was purified by DNase I digestion and ethanol precipitation, and equimolar amounts of sense and antisense RNAs were annealed to generate double-stranded RNA (dsRNA). When cytosolic GPXL-dsRNA was prepared, it was preserved at −80 °C for further use. NEPA21 (NEPA GENE, Chiba, Japan) was used for the electroporation of the prepared cytosolic GPXL-dsRNA in two-day-old *E. gracilis* cells under the conditions mentioned in Appendix A. A 100 µL electroporation solution containing 15 µg of GPXL-dsRNA, ≈1 × 10^6^ cells, with KH media to adjust total volume to 100 µL, was prepared and subsequently taken into a cuvette cell with a 0.2 cm gap. Afterward, electroporation was performed using NEPA21. Then, the 100 µL electroporation solution was taken in 10 mL KH media for continuous incubation at 26 °C with 120 rpm under dark for 5 days. In addition to cytosolic *EgGPXL*s, two other cytosolic peroxidases, *EgPrx-1* and *EgPrx-4*, were silenced simultaneously using the same procedure mentioned above. Finally, suppression of the target was confirmed by semi-quantitative reverse transcription (RT)-PCR using specific primer sets (Appendix A). In addition, elongation factor 1 alpha (EF1α; Acc NO. X16890) was amplified by RT-PCR for normalization using the primer set mentioned in Appendix A. 

### 2.8. Spot Assay

Genetic suppression of cytosolic *EgGPXL*s in *E. gracilis* was assessed using a growth-based analysis called the spot assay. A defined number of cells (6 × 10^6^, 3 × 10^6^, 1.5 × 10^6^, 7.5 × 10^5^, and 3.75 × 10^5^) from mutant and wild-type *E. gracilis* were spotted on KH and CM agar plates for heterotrophic and autotrophic growth conditions, respectively. Further, the plates were taken under normal-light and high-light conditions at 26 °C for 1 week. For each condition, 100 μmol photons m^−2^ s^−1^ and 350 μmol photons m^−2^ s^−1^ were set, respectively, in the growth chamber. The final conditions were set to induce oxidative stress in *E. gracilis*. 

### 2.9. Intracellular H_2_O_2_ Detection

Intracellular production of H_2_O_2_ was measured in wild-type and mutant cell lines using H_2_O_2_ specific fluorescent probe known as BES-H_2_O_2_-Ac (derivative of BES (benzenesulfonyl) (Wako, Japan)). After collecting 200 µL *Euglena* cells, BES-H_2_O_2_-Ac was added to all the cell lines at a final concentration of 10 μM. Each cell line was microcentrifuged at 1000× *g* for 5 min after incubation in the dark for 30 min at room temperature. The supernatant was then removed. Consequently, the fluorescence intensity of all cell lines was determined using a fluorescence detection system (Corona-SH 9000; Higashi-Ishikawa, Japan). 

### 2.10. Glutathione and Chlorophyll Determination

Glutathione and Chlorophyll measurements were performed in wild-type and mutant cell lines as described previously [31,32], without any further modification. 

### 2.11. Statistical Analysis

Statistical analysis was conducted using Dunnett’s *t*-test and ANOVA (*p* < 0.05). The calculations were performed using at least three independent biological replicates.

## 3. Results and Discussion

### 3.1. Molecular Identification and Structural Analysis of EgGPXLs 

*Euglena* GPXL gene sequences were searched in our previously published RNA-seq data, and the tblastn tool was used to acquire the complete cDNA sequence of EgGPXLs from published transcriptomic data [26]. Full-length cDNAs for EgGPXLs were obtained as a spliced leader sequence (TATTTTTTTTCG) found at the 5′-end of the obtained sequences, except for EgGPXL-3. *Euglena* transcripts usually contain this spliced leader sequence, which is produced through a process called trans-splicing, a phenomenon found in *E. gracilis* and other organisms [33,34]. Among these three cytosolic EgGPXLs, EgGPXL-3 and EgGPXL-4 were plant type, and EgGPXL-2 was animal type, with respect to evolutionary relationships within other GPX from different domains [27]. Homology analysis revealed that EgGPXL-1, EgGPXL-3, and EgGPXL-4 were highly identical (36.0–51.2%), whereas EgGPXL-2 showed the lowest identity (21.5–30.8%) to the remaining EgGPXLs [27]. Properties of EgGPXLs and GPXs from other organisms were explored using ClustalW-based multiple sequence alignment analysis. Similar to other GPX, these cytosolic EgGPXLs also contain three types of GPX signature motifs, as illustrated in Figure 1 [35,36]. Some conserved amino acid residues found across all EgGPXLs may be responsible for the redox center, as mentioned in previous studies [34,35]. The conserved amino acid residues were found to be EgGPXL-2 (Asn-40, Ala-41, Pro-73, Cys-74, Gln-77, Phe-75, Glu-85, Gly-110, Try-130, Asn-131, Phe-132, Lys-135, Gly-142), EgGPXL-3 (Asn-12, Ala-13, Pro-43, Cys-46, Gln-48, Phe-49, Glu-53, Gly-82, Try-107, Asn-108, Phe-109, Lys-111, Gly-118), and EgGPXL-4 (Asn-31, Ala-33, Pro-63, Cys-64, Gln-66, Phe-67, Glu-71, Gly-100, Try-125, Asn-126, Phe-127, Lys-129, Gly-136). In the GPX signature motif 1, selenocysteine, the main catalytic residue of GPXs in many prokaryotic and eukaryotic organisms except plants, certain insects, nematodes, and various protists, has been evolutionarily replaced by cysteine (Figure 1) [10]. A similar result was found in early branching fungal phyla, where fungal species used an altered mechanism of Sec insertion due to the absence of Sec recoding signals (SECIS elements) [12]. Furthermore, this replacement in EgGXPLs might help resist some critical situations created under specific environmental conditions, similar to [NiFe]-hydrogenase, where selenocysteine is replaced with cysteine and confers resistance to oxygen tolerance [37].

### 3.2. Characterization of Primary Structure of EgGPXLs

The cDNA sequences obtained were 737, 857, and 787 bp for EgGPXL-2 (GC content-55.22%), EgGPXL-3 (GC content-66.51%), and EgGPXL-4 (GC content-60.74%), respectively, which translated into 173, 144, and 159 amino acid residues, respectively. Physicochemical parameters, including the molecular weight (MW), isoelectronic points (pI), and grand average of hydropathy (GRAVY), were predicted using the PROTPARAM tool (http://web.expasy.org/protparam/, accessed on 10 April 2024). The deduced molecular weight, isoelectric pH (pI), and GRAVY value of EgGPXL-2, EgGPXL-3, and EgGPXL-4 were 16 to 20 kDa, 6.0 to 8.34, and −0.106 to −0.419, respectively. The predicted molecular weights were also in accordance with the SDS-PAGE analysis of the individual recombinant proteins (Figure 2). All EgGPXLs were expected to be localized within the cytosol, as no signal peptide was found in any of them, according to in silico tools such as TargetP (https://services.healthtech.dtu.dk/services/TargetP-2.0/, accessed on 10 April 2024) and TMHMM (https://services.healthtech.dtu.dk/services/TMHMM-2.0/, accessed on 10 April 2024) (Appendix A). In addition, no signal peptide was found in the SMART domain analysis, which also explored the presence of the GSHPx domain (Glutathione Peroxidase Domain) in all three EgGPXLs, which was consistent with previous studies (Appendix A) [10,36]. *Euglena*-derived peroxidases located in the mitochondria, chloroplasts, or other organelles usually contain signal peptides similar to other peroxidases obtained from various organisms [24,38]. In our previous study, we demonstrated that chloroplastic and mitochondrial EgPrxs contained signal peptides, whereas cytosolic EgPrxs lacked signal peptides. 

Additionally, the mitochondrial EgPrx-3 contains an LRR motif, which is a characteristic motif of *Euglenozoa* mitochondrial proteins [24]. Additionally, *Euglena* derived other peroxidases, such as Prx-1, Prx-4, and APX, localized in the cytosol and lacked any type of signal peptide [23,24]. Furthermore, EgGPXL-1 contained a transmembrane domain at the 5’-end, indicating its potential localization within the chloroplast, similar to other signal peptide-oriented peroxidase enzymes [28]. Therefore, in silico analysis and previous studies indicated the potential localization of EgGPXLs in the cytosol. 

Post-translational modification (PTM) mechanisms are crucial influencers of cellular signaling and the chemical properties of target proteins. Proteins modified through these mechanisms alter the spatial structure, stability, folding characteristics, subcellular localization, and biological functions of the target proteins [39]. In these cytosolic EgGPXLs, many PTM sites were predicted using the MusiteDeep PTM prediction (https://www.musite.net/ (accessed on 16 May 2024)) and Deep Nitro (http://deepnitro.renlab.org/index.htm (accessed on 16 May 2024)) online servers (Appendix A). Some of the predicted PTM sites were also detected in the motifs obtained using the MEME online server (https://meme-suite.org/meme/ (accessed on 16 May 2024)). Five motifs were predicted in EgGPXL-2, whereas motif-3 (Phe-72 to Tyr-117) contained two PTM sites (methylation at Arg-93 and nitration at Try-79), and the remaining motifs contained only one PTM site except motif-4 (Appendix A). Moreover, the PTM site in motif-2 was found to be Cys-45, which is substituted with selenocysteine and is widely conserved in almost all domains of GPXs, with some exceptions [12,35]. In the case of EgGPXL-3, three motifs were predicted, and cumulatively, 11 PTM sites were discovered in motif-2 (Phe-44 to Tyr-89) and motif-3 (His-91 to Glu-138) (Appendix A). One palmitoylation site (Cys-46) was predicted at motif-2, which is a cysteine residue universally conserved across all GPX domains and one of the active sites of EgGPXL-3. Furthermore, one SUMOylation (Lys-51) and one acetylation (Lys-77) site were predicted in this motif. In addition, two phosphorylation sites (Ser-125, 132), two glycosylation sites (Asn-108 and Thr-128), one SUMOylation site (Lys-93), and one acetylation site (Lys-111) were predicted to be present in motif-3. Remarkably, another PTM site, Lys-134, was predicted to be involved in three PTM named ubiquitination, SUMOylation, and methylation in EgGPXL-3. EgGPXL-4 contains four motifs, and several PTM sites were predicted for motif-1 (Met-1 to Phe-7), motif-3 (Phe 62 to Trp-107), and motif-4 (His-109 to Glu-156). The highest number of PTM sites was found in motif-4, namely Asp-126, Lys-129, Pro-148, and Ser-149,150, which were anticipated to be involved in glycosylation, acetylation, hydroxylation, and phosphorylation, respectively. In addition, two other amino acid residues are predicted to be involved in double PTM, namely I) Lys-134, which is involved in acetylation and ubiquitination, and II) Thr-146, which is involved in phosphorylation and glycosylation. Other PTM sites were also found across EgGPXL sequences (Appendix A). Previously, *S*-nitrosylation and tyrosine nitration manipulated the activity of some peroxidase enzymes involved in the AsA-GSH cycle in plants, like monodehydroascorbate reductase, dehydroascorbate reductase, glutathione reductase, and APX [40,41,42,43]. Furthermore, cold-stress resistance was increased in chrysanthemum through a post-translational modification of the GPX protein, called decrotonylation, which elevated GPX activity under this stress condition [44]. In addition, some PTM had been reported in mammalian GPX-4, such as phosphorylation, *N*-linked glycosylation, and SUMOylation, which were responsible for sperm maturation, differentiation, and the inhibition of lipid peroxidation, respectively [45]. Recently, PTM in mammalian GPX-4, such as ubiquitination, succination, and alkylation, have become attractive therapeutic tools for treating ferroptosis-related diseases [45]. In vivo and in vitro analyses of catalase obtained from human erythrocytes demonstrated that this enzyme was sensitive to some PTMs such as *S*-nitrosation, *O*-glycosylation, Tyr-nitration, and *S*-glutathionylation, whereas *S*-nitrosation inhibits catalase activity in children with obesity and insulin resistance [46]. In view of the above discussion, the predicted PTM sites may play a crucial role in *Euglena* under normal or stressful conditions. 

### 3.3. Substrates and Electron Donor Profiling of rEgGPXLs 

Hexahistidine-tagged rEgGPXL-2, -3, and -4 were expressed in the *E. coli* BL21 expression system. Subsequently, nickel (II) chelate affinity chromatography was employed to purify the target proteins, and confirmation was achieved by SDS-PAGE analysis (Figure 2). Unfortunately, rEgGPXL-3 could not be purified by nickel (II) chelate affinity chromatography because of the presence of inclusion bodies. The SDS-PAGE analysis indicated that the proteins exhibited molecular weights ranging from 18 kDa to 20 kDa, consistent with the predicted molecular weights. Previously, one of the molecularly uncharacterized cytosolic glutathione peroxidase homologs of *Euglena* used GSH as the electron donor, and GSH-dependent peroxidase activity was initially assessed for both purified rEgGPXL-2 and rEgGPXL-4 following a previously described methodology [47]. However, no GSH-dependent peroxidase activity was detected. Notably, rEgGPXLs exhibited peroxidase activity when Trx was used as an electron donor. This might have occurred due to the higher affinity of EgGPXLs towards Trx, as well as the achievement of an optimal redox potential during interaction with Trx [2,17]. Additionally, the substitution of selenocysteine with cysteine might be another possible reason for the higher affinity towards the Trx system over the GSH system, which is also assumed to regenerate the Trx system in plants during redoxtasis [5,10]. Similarly, various plant GPXs (*Lycopersicon esculentum*, *Helianthus annuus*, *Arabidopsis thaliana*, and Chinese cabbage) and *Euglena* Prxs used Trx instead of GSH as an electron donor [5,24,38,48,49,50]. 

Furthermore, GPX-5 from the microalga *C. reinhardtii* used the same electron donor as rEgGPXLs [51]. The specific peroxidase activity for rEgGPXL-2 using H_2_O_2_ and *t*-BOOH as substrates were determined as 1.3 µmol/min/mg protein and 0.79 µmol/min/mg protein, respectively. As for rEgGPXL-4, its specific peroxidase activity was measured at 4.9 µmol/min/mg protein and 3.5 µmol/min/mg protein, respectively, for the same substrate. Substrate profiling of rEgGPXLs indicated that rEgGPXL-2 and rEgGPXL-4 reduced *t*-BOOH more efficiently than they reduced H_2_O_2_. Additionally, the specific activity of rEgGPXL-4 for *t*-BOOH revealed that it had the maximum catalytic efficiency during catalysis with *t*-BOOH compared to other *Euglena* peroxides, except for cytosolic EgAPX [23,24] (Table 1). These two substrates were used in a concentration-dependent manner for peroxidase activity to determine other kinetic parameters, such as *K*m, *k*cat, *k*cat/*K*m, of these two cytosolic rEgGPXLs (Table 1).

During H_2_O_2_ and *t*-BOOH-dependent peroxidase activity, a hyperbolic graph was observed, which indicated non-allosteric enzymatic regulation or Michaelis–Menten-type enzymatic regulation of these rEgGPXLs (Figure 3). A similar pattern was observed when Trx-dependent peroxidase activity was assessed for both substrates (Figure 4). Moreover, the Vmax values of rEgGPXLs indicated faster catalytic efficiency than GPXs from other green algae, such as *Chlorella* sp. NJ-18 and *C. reinhardtii*, which showed 18.6 to 317.8 nmol/min/mg protein (Table 1) [52]. The *k*cat/*K*m ration of rEgGPXLs pointed out lower efficiency in product conversion from substrate compared to other *Euglena* peroxides as well as selenocysteine containing human hGPX-3 but almost similar to *Arabidopsis* GPX-1, -2, -5, and -6 (converted 4.9 × 10^3^, 4.5 × 10^3^, 3.1 × 10^3^, and 6.1 × 10^3^ M of H_2_O_2_ into product per second) [24,50,53]. 

A clear pH-dependent pattern was observed for the rEgGPXL peroxidase activity (Figure 5). rEgGPXLs showed the highest peroxidase activity at pH 6; beyond pH 6, a sharp decline in peroxidase activity was observed (Figure 5). This result implies that cytosolic EgGPXLs possess maximum catalytic activity within a slightly acidic to neutral pH range. The present result is in accordance with that of other GPX and GST enzymes from *Lactobacillus plantarum*, which showed maximum peroxidase activity at pH 6 within a pH range of 4–7 [54]. 

Furthermore, a significant reduction in peroxidase activity was observed when rEgGPXLs catalyzed the enzyme at pH 8.0 and above. In contrast, the GPX from the Antarctic psychrotrophic bacterium *Pseudoalteromonas* sp. ANT506 showed maximum peroxidase at alkaline pH (pH 9.0) and conserved almost 45% peroxidase within the pH range of 7.0 to 10.0 [55]. Enzymes lose peroxidase activity at higher pH for several reasons, such as changes in the enzyme’s active site, structural stability, and the ionization state of critical catalytic residues, which may also be possible reasons for these rEgGPXLs.

### 3.4. Impacts of GPXL Suppression on E. gracilis

Functional analysis of the EgGPXLs was conducted using RNAi. EgGPXL-derived tailored dsRNA molecules were produced and introduced into *E. gracilis* cells via electroporation. Consequently, the expression of *EgGPXL*s was silenced, which was later confirmed using RT-PCR (Figure 6). As a suppression of cytosolic peroxidase homologs named *EgPrx-1* and *EgPrx-4* was found to be critical for *E. gracilis*, these two homologs were also silenced in parallel to check the synergistic effect of all these cytosolic homologs. In this study, three types of mutant cell lines were produced through RNAi, namely, double-mutant EgPrx-1/EgPrx-4; triple-mutant EgGPXL-2/EgGPXL-3/EgGPXL-4; quintuple-mutant EgPrx-1/EgPrx-4/EgGPXL-2/EgGPXL-3/EgGPXL-4 where *EgPrx-*1/*EgPrx-4*, *EgGPXL-2*/*EgGPXL-3*/*EgGPXL-4*, and *EgPrx-1*/*EgPrx-4*/*EgGPXL-2*/*EgGPXL-3*/*EgGPXL-4* genes were silenced, respectively. 

To assess the effect of these cytosolic peroxidases on *E. gracilis*, a spot assay was performed after confirming the silencing of these homologs. According to the spot assay, a triple-mutant cell line (ΔGPXL-2/-3/-4) was unable to induce any noticeable critical condition in *E. gracilis* under normal- (100 μmol photons m^−2^ s^−1^) and high-light (350 μmol photons m^−2^ s^−1^) conditions, whereas the growth of a double-mutant cell line (ΔPrx-1/-4) was arrested in the same condition as found previously [24] (Figure 7). Similar to the triple-mutant cell line, the suppression of cytosolic EgAPX did not change the growth of *Euglena* but increased the level of H_2_O_2_ [23]. No noticeable critical condition in the triple-mutant cell line indicated the involvement of compensatory or redundant mechanisms in *Euglena*, which helped this cell line restore to normal conditions. In addition, several other factors may play crucial roles in restoring the normal conditions of the triple-mutant cell line. First, cytosolic EgGPXLs may be regulated or closely interconnected within a sophisticated redox networking system; therefore, no critical conditions may be induced without further alteration of other genes in this system. Second, the H_2_O_2_ produced by the silencing of cytosolic EgGPXLs under normal or abiotic stress conditions might be reduced only by the cytosol-centric APX of *E. gracilis* [23]. 

In addition, the photosynthetic ability of *E. gracilis* is resistant to high concentrations of H_2_O_2_ (up to 1 mM) compared *Arabidopsis* because of the presence of H_2_O_2_-insensitive enzymes, including fructose-1,6-bisphosphatase, sedoheltulose-1,7-bisphosphatase, NADP^+^-glyceraldehyde-3-phospahte dehydrogenase, and ribulose-5-phosphate kinase in the Calvin cycle, which help it to endure high concentrations of H_2_O_2_ [56]. Similar to plants, *E. gracilis* has non-enzymatic antioxidant molecules, such as ascorbate and glutathione, which might contribute to the scavenging of ROS generated after the suppression of cytosolic *EgGPXL*s [30]. Furthermore, H_2_O_2_ transporters, such as aquaporin, may be involved in the efflux of increased H_2_O_2_ from the cytosol into the extracellular space [22,57]. Interestingly, the downregulation of *EgPrx-1* and *EgPrx-4* did not show any critical phenotype in the quintuple-mutant cell line, similar to the double-mutant cell line, indicating that repression of *EgGPXL-2*/*EgGPXL-3*/*EgGPXL-4* restored normal conditions in this cell line (Figure 7). Furthermore, the number of cells in the quintuple-mutant line was lower than that in the wild-type but higher than that in the double-mutant line, while no changes were found between the triple-mutant and wild-type cell numbers, supporting the spot assay analysis (Figure 8). The existence of the trypanothione system has already been proven in *E. gracilis*, and the present results reinforce the existence of another antioxidant system, such as trypanothione/tryparedoxin, usually found in the sister group of *Euglena* called kinetoplastida, to compensate for stress conditions in triple- and quintuple-mutant cell lines [21]. Previously, normal conditions were restored in tomato res mutants (cell structure was restored by salinity), causing chlorosis and developmental alterations in tomatoes under salt stress conditions through the induction of chloroplast-targeted DEAD box proteins [58]. Similar to *Euglena* triple- and quintuple-mutant cell lines, the suppression of *GPX-1*, *-3*, and *-6* in *Arabidopsis* did not induce any changes in shoot biomass, leaf pigment, or leaf number [59]. Accordingly, impairment of some rice APX like cytosolic *OsAPX-1* and *OsAPX-2*, peroxisomal *OsAPX-3* and *OsAPX-4*, stromal *OsAPX-7*, and *Arabidopsis* thylakoid membrane-attached APX (tAPX), failed to provoke any significant changes in photosynthesis, growth, as well as development under normal conditions [60,61,62,63,64]. Notably, rice becomes resistant to different abiotic stresses, such as salt, heat, and drought, because of the suppression of cytosolic *OsAPX-1*, *OsAPX-2*, and stromal *OsAPX-7* [60,61]. In contrast to earlier studies, the causative agent of listeriosis, *Listeria monocytogenes*, acquired resistance and pathogenicity after the suppression of a putative GPX homolog (lmo0983) [64].

APX- and Trx-dependent peroxidase activities were also assessed in all mutant cell lines (Figure 9). Regarding APX activity, no significant changes were observed in any of the cell lines, indicating that APX expression or transcriptional regulation was not triggered after silencing cytosolic *EgGPXL*s/*EgPrx*s (Figure 9A). 

Similarly, the downregulation of peroxisomal APX in rice, stromal *OsAPX-7* in rice, and cytosolic *EgAPX*, *EgPrx1/4*, chloroplastic *EgPrx-2*, and mitochondrial *EgPrx-3* in *Euglena* did not significantly change APX activity significantly [23,24,61]. Furthermore, Trx-dependent peroxidase activity was unchanged in triple- and quintuple-mutant cell lines, whereas it was significantly reduced in the double-mutant cell line, as found previously (Figure 9B) [24]. As chloroplastic EgPrx-2 and EgGPXL-1, as well as mitochondrial EgPrx-3, were found in *E. gracilis*, triple- and quintuple-mutant cell lines may restore Trx-dependent peroxidase activity.

A quantitative measurement of chlorophyll and glutathione content determines the physical state of photosynthetic organisms; therefore, it was measured in all cell lines. Unfortunately, there was no significant difference in chlorophyll and glutathione content in any mutant cell line compared to the wild-type (Figure 10 and Figure 11). As *Euglena* contains other peroxidases in different cellular compartments, such as cytosolic APX, chloroplastic EgPrx-2, EgGPXL-1, and mitochondrial EgPrx-3, the produced H_2_O_2_, after silencing these cytosolic peroxidases, might be reduced by peroxidase enzymes. Consequently, it did not hamper the chlorophyll content in the mutant cell lines (Figure 10). In addition, it also contained GR, responsible for the regeneration of GSH, which might ensure a continuous supply of GSH in *Euglena* with the help of NADPH; hence, the GSH redox state remained unchanged in mutant cell lines after silencing cytosolic peroxidase (Figure 11) [21]. 

Additionally, quantitative determination of H_2_O_2_ was performed in *Euglena* mutant cell lines using the chemical fluorescent probe BES-H_2_O_2_-Ac. Similar to GSH and chlorophyll content, the fluorescence intensity of all mutant cell lines was similar to that of the wild-type (Figure 12). According to a previous study, the suppression of EgAPX increases the accumulation of H_2_O_2_ in *E. gracilis*, indicating the physiological importance of APX during ROS homeostasis [16]. Additionally, it suggested a higher affinity of cytosolic EgAPX towards H_2_O_2_, which was also revealed by its much higher specific activity compared to any type of peroxidase enzyme mentioned previously in *E. gracilis* [23,24]. In contrast, the specific activities of EgGPXLs and EgPrxs were much lower than those of EgAPX; therefore, the suppression of these homologs did not induce any significant change in the amount of H_2_O_2_ compared to the wild-type (Figure 12). 

Eventually, the produced H_2_O_2_ under high-light conditions after the suppression of these cytosolic *EgGPXL*s and *EgPrx*s might be reduced only by APX in *Euglena*. Although further studies are needed, cytosolic EgGPXLs and EgPrxs may be limited in their involvement in H_2_O_2_ metabolism and may contribute to the metabolism of other lipid peroxides and/or thiol compound homeostasis in *E. gracilis*.

## 4. Conclusions

In conclusion, the suppression of cytosolic *EgGPXL*s indicated the potential resistance efficiency of *Euglena* towards ROS, as well as the association of compensatory mechanisms to remain unchanged under photo-oxidative stress conditions. The restoration of cytosolic *EgPrx-1/4* from critical conditions highlights its potential role as a negative regulator of cytosolic *EgGPXL*s in *E. gracilis*. Finally, comprehensive genetic manipulation, such as the silencing of peroxidase enzymes in various combinations, is required to understand the complex antioxidant networking system in *E. gracilis*. 

## Figures and Tables

**Figure 1 biomolecules-14-00765-f001:**
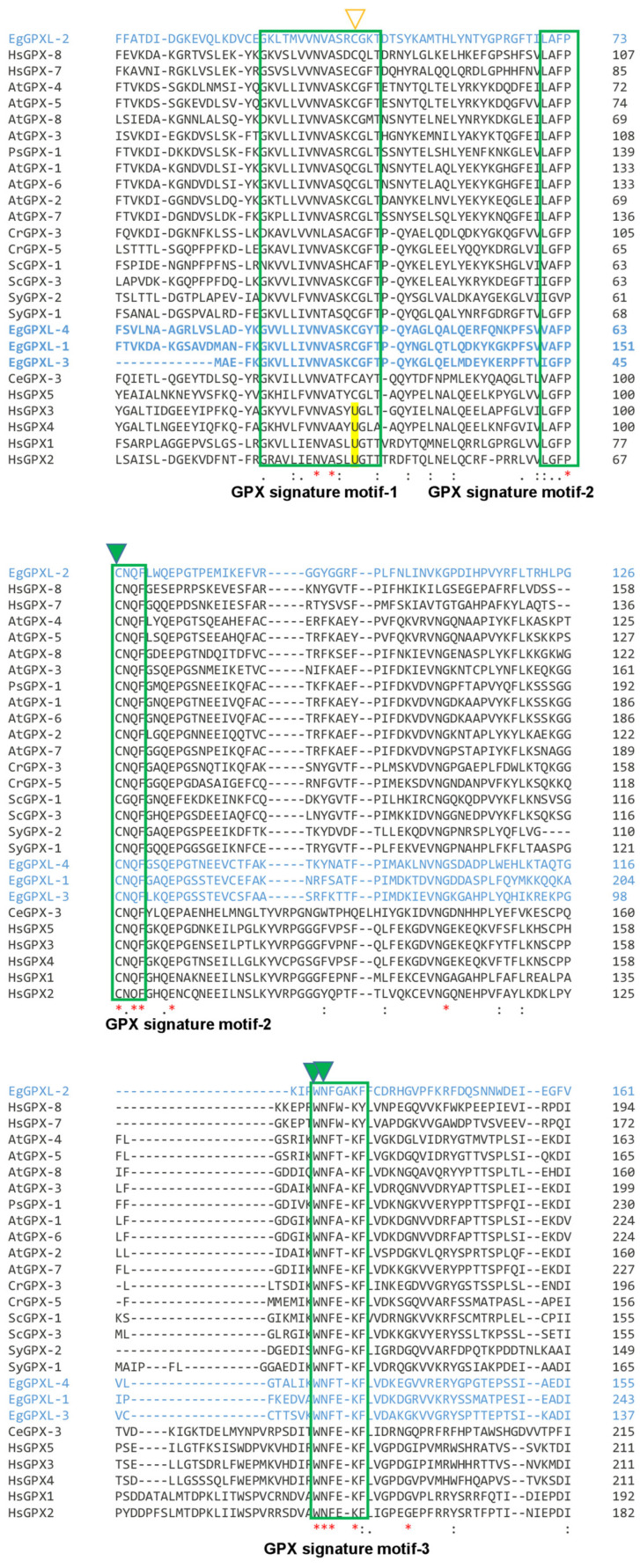
Sequence analysis of GPXL proteins derived from *Euglena* as well as other organisms. Three GPX signature motifs were found ubiquitously across all domains of GPXLs distinguished by green boxes. A basic (Asp), a nonpolar (Try), and a thiol (Cys) amino acid, denoted by green triangle, might be responsible for catalysis, highly conserved in almost all the GPXLs from various domains. In contrast, plant GPXs and EgGPXLs contained two cysteine residues, which were indicated in orange triangle and green triangle. Additionally, amino acid residues demonstrating identical characteristics throughout all sequences were marked with a red asterisk below. Furthermore, residues displaying similar traits were indicated with paired dots beneath the sequences, whereas those with a degree of resemblance were identified with a single dot. EgGPXLs were highlighted with blue color. Selenocysteine (U), found in many prokaryotes and eukaryotes except plants, certain insects, nematodes, and various protists, was replaced evolutionarily by cysteine in remaining GPXs (highlighted in yellow color).

**Figure 2 biomolecules-14-00765-f002:**
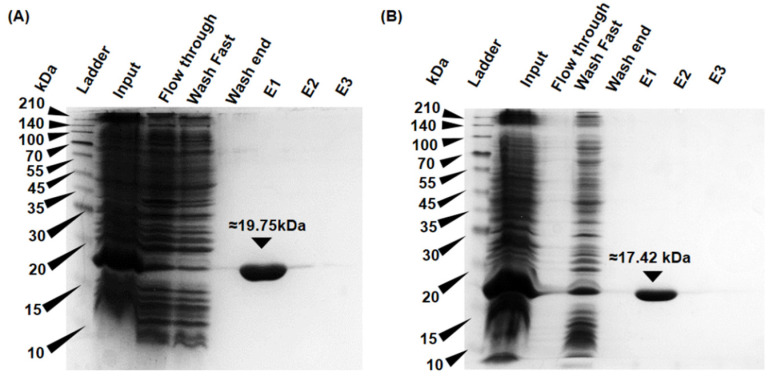
SDS-PAGE-based expression and purification analysis of 6X-His tagged rEgGPXL-2 (**A**) and rEgGPXL-4 (**B**). The rEgGPXL-2 and -4 were taken in TALON Metal Affinity resin for purification after expression in *E. coli* Bl21. Eight types of samples were run in this electrophoresis system which were input from Lane 1 to 7. Sample after expression, flow-through (first fraction in purification), sample obtained in first washing step, and sample obtained after last washing step were loaded in Lanes 2 to 4, correspondingly. Through three consecutive elutions loaded in Lanes 5 to 7, respectively, we tried to find purified proteins. After separation of proteins through SDS/PAGE, it was visualized with Coomassie Brilliant Blue. M denoted the molecular weight marker.

**Figure 3 biomolecules-14-00765-f003:**
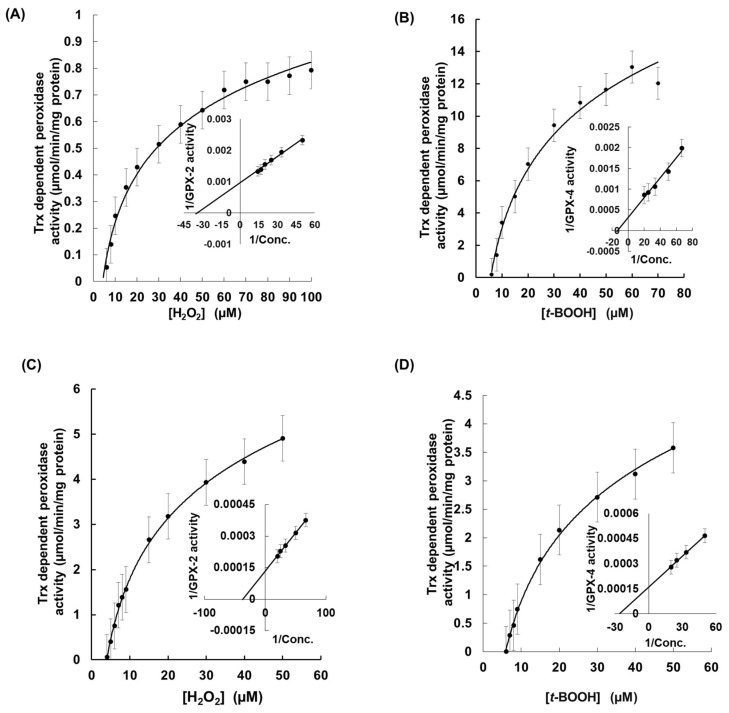
The kinetic behavior of cytosolic rEgGPXLs. Hyperbolic graph was obtained for rEgGPXL-2 and rEgGPXL-4 when H_2_O_2_ (**A**,**B**) and *t*-BOOH (**C**,**D**)-dependent peroxidase activity was performed for these cytosolic proteins, respectively. (**A**,**B**) denoted H_2_O_2_ and *t*-BOOH-dependent peroxidase activity for rEgGPXL-2, respectively; similarly, (**C**,**D**) denoted it for rEgGPXL-4. The Lineweaver–Burk plots (inset) were constructed through best fit lines. The values represented the mean average of three consecutive experimental runs.

**Figure 4 biomolecules-14-00765-f004:**
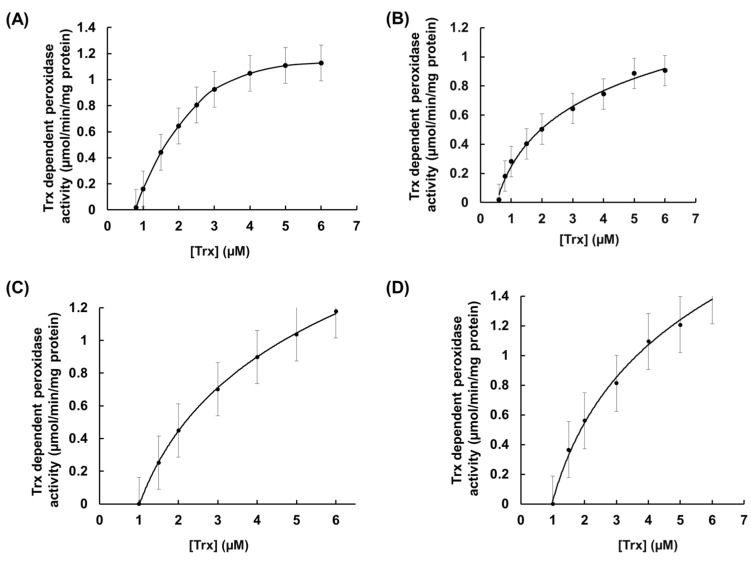
Thioredoxin (Trx)-dependent peroxidase activity of cytosolic rEgGPXLs. (**A**,**B**) indicated Trx-dependent peroxidase activity using H_2_O_2_ and *t*-BOOH as substrates for rEgGPXL-2, respectively. Similarly, (**C**,**D**) denoted it for rEgGPXL-4. The values represented the mean average of three consecutive experimental runs.

**Figure 5 biomolecules-14-00765-f005:**
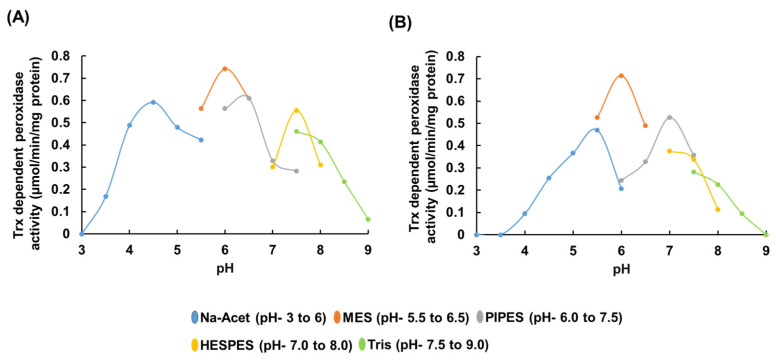
pH dependency of cytosolic rEgGPXLs. This result implied optimum activity of rEgGPXL-2 (**A**) and rEgGPXL-4 (**B**) from slightly acidic to neutral pH. After pH 8.0 and above, a significant reduction of peroxidase activity was noted in both enzymes. The values represented the mean average of three consecutive experimental runs.

**Figure 6 biomolecules-14-00765-f006:**
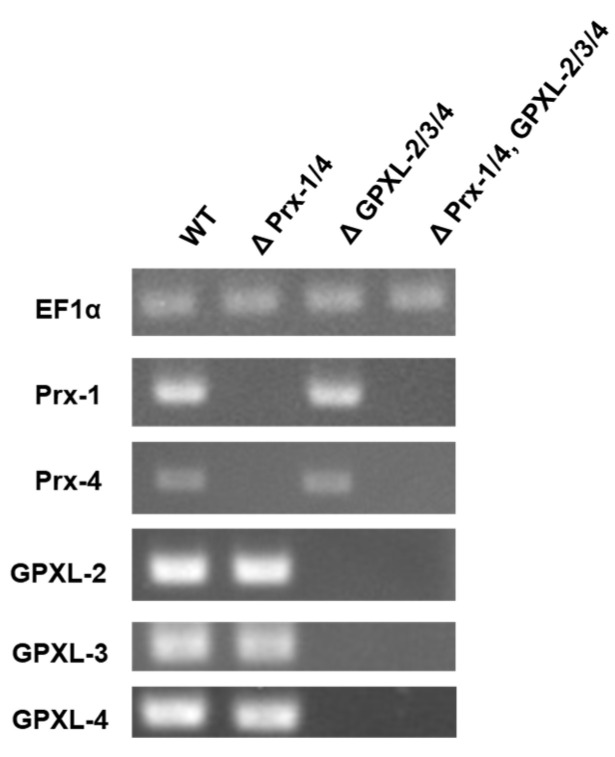
RT-PCR-based confirmation of cytosolic *EgGPXL*s silencing. In addition to cytosolic *EgGPXL*s, another two cytosolic peroxidases, *EgPrx-1* and *EgPrx-4*, were also silenced for investigation of synergistic effect of these cytosolic peroxidases. Therefore, quantitative analysis of gene expression was conducted for five types of peroxidase genes named *EgGPXL-2*, *EgGPXL-3*, *EgGPXL-4*, *EgPrx-1*, *EgPrx-4,* and one housekeeping gene, *EF1α*, through RT-PCR. PCR amplification of *EgGPXL-2*, *EgGPXL-3*, *EgGPXL-4*, *EgPrx-1*, *EgPrx-4,* and *EF1α* was conducted using cDNA templates obtained from the WT and mutant cell lines. Lane 1 contained WT (control, electroporation without dsRNA); Lanes 2 to 4 contained samples from three mutant cell lines, ΔPrx-1/4, ΔGPXL-2/-3/4, and ΔPrx-1/4, GPXL-2/3/4, respectively. The result was confirmed after repeating three consecutive experimental runs.

**Figure 7 biomolecules-14-00765-f007:**
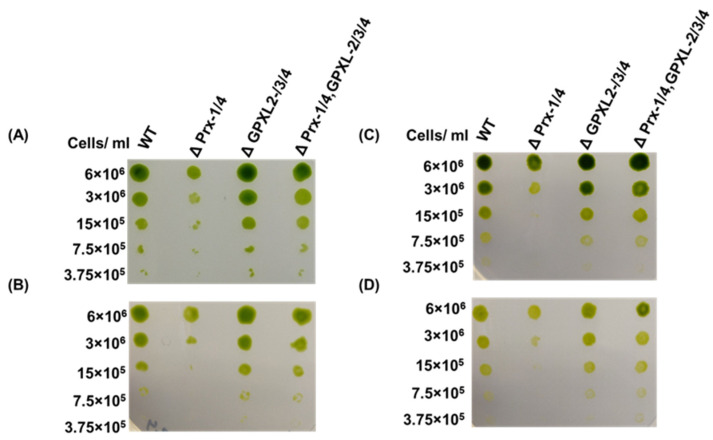
Observation of phenotypic change after suppression of cytosolic *EgGPXL*s. In addition to cytosolic *EgGPXL*s, another two cytosolic peroxidases, *EgPrx-1* and *EgPrx-4*, were also silenced for investigation of synergistic effect of these cytosolic peroxidases. Wild-type and mutant cell lines were applied as spots onto KH plates used for heterotrophic growth conditions (**A**,**B**) and onto CM plates designated for autotrophic growth conditions (**C**,**D**). The cell quantities spotted were 6 × 10^6^, 3 × 10^6^, 15 × 10^5^, 7.5 × 10^5^, and 3.75 × 10^5^ in both agar plates. Subsequently, the plates were taken to a 7-day incubation period under normal growth conditions (26 °C with 100 μmol photons m^−2^ s^−1^) (**A**,**C**) and high-light conditions (300 μmol photons m^−2^ s^−1^) (**B**,**D**) to induce oxidative stress in the cell lines. ΔPrx-1/4, ΔGPXL-2/3/4, and ΔPrx-1/4, GPXL-2/3/4 corresponded to the cell lines where *Prx-1*/*Prx-4*, *GPXL-2*/*GPXL-3*/*GPXL-4*, and *Prx-1*/*Prx-4*, *GPXL-2*/*GPXL-3*/*GPXL-4* genes were suppressed, respectively. The result was confirmed after performing three consecutive replications.

**Figure 8 biomolecules-14-00765-f008:**
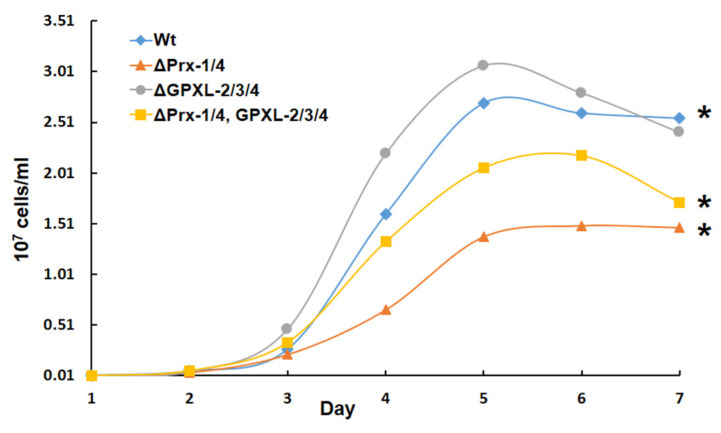
Growth rates of the control and each KD cell. Cells were grown heterotrophically under dark at 26 °C for 8 days. ΔPrx-1/4, ΔGPXL-2/3/4, and ΔPrx-1/4, GPXL-2/3/4 corresponded to the cell lines where *Prx-1*/*Prx-4*, *GPXL-2*/*GPXL-3*/*GPXL-4*, and *Prx-1*/*Prx-4*, *GPXL-2*/*GPXL-3*/*GPXL-4* genes were suppressed, respectively. Values are the mean ± SD (*n* = 3). Values with asterisks were significantly different, according to the ANOVA (*p* < 0.05).

**Figure 9 biomolecules-14-00765-f009:**
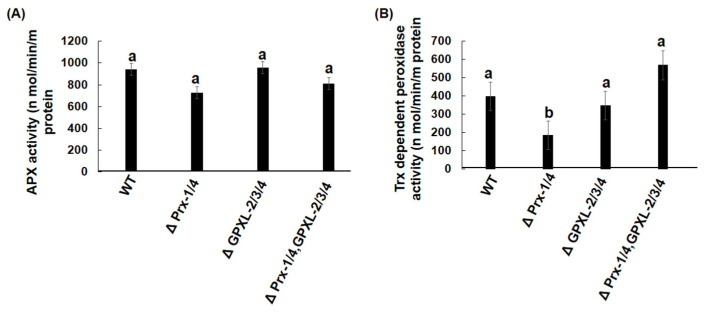
Effect of suppression of EgGPXLs on APX and Trx dependent peroxidase activity in the wild-type (WT) and mutant cell lines. APX (**A**) and Trx-dependent peroxidase (**B**) were checked in extracts derived from both the wild-type cells and each individual mutant cell line, respectively. ΔPrx-1/4, ΔGPXL-2/3/4, and ΔPrx-1/4, GPXL-2/3/4 corresponded to the cell lines where *Prx-1*/*Prx-4*, *GPXL-2*/*GPXL-3*/*GPXL-4*, and *Prx-1*/*Prx-4*, *GPXL-2*/*GPXL-3*/*GPXL-4* genes were suppressed, respectively. Values are the mean ± SD (*n* = 3). Values with same letter indicated no significant difference in enzyme activity, while those with different letters had significant differences in enzyme activity in all cell lines according to Dunnet *t*-test (*p* < 0.05).

**Figure 10 biomolecules-14-00765-f010:**
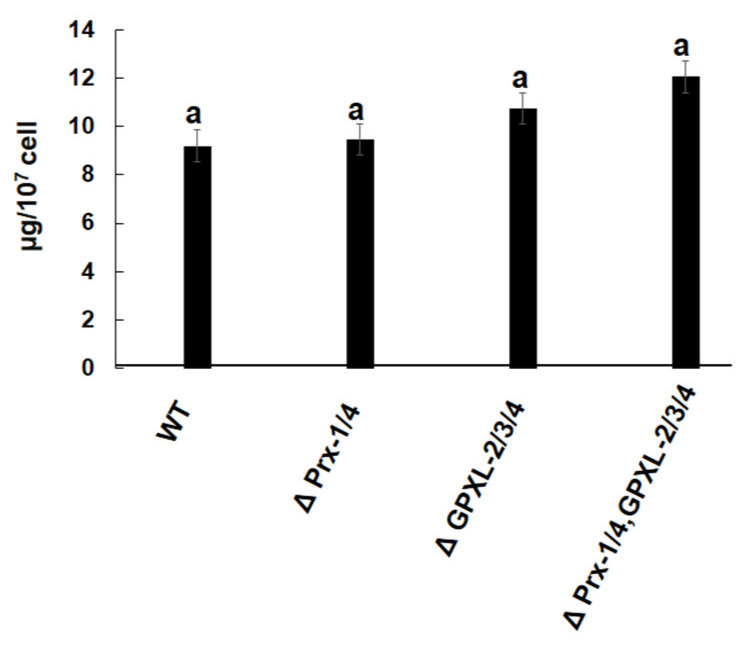
Assessment of chlorophyll content in mutants and wild-type cell line of *Euglena*. ΔPrx-1/4, ΔGPXL-2/3/4, and ΔPrx-1/4, GPXL-2/3/4 corresponded to the cell lines where *Prx-1*/*Prx-4*, *GPXL-2*/*GPXL-3*/*GPXL-4*, and *Prx-1*/*Prx-4*, *GPXL-2*/*GPXL-3*/*GPXL-4* genes were suppressed, respectively. Chlorophyll was measured for both chlorophyll a and b, according to previously mentioned protocol, without further modification [32]. Values with same letter indicated no significant difference in chlorophyll content in mutant cell lines compared to wild type, according to Dunnet *t*-test (*p* < 0.05).

**Figure 11 biomolecules-14-00765-f011:**
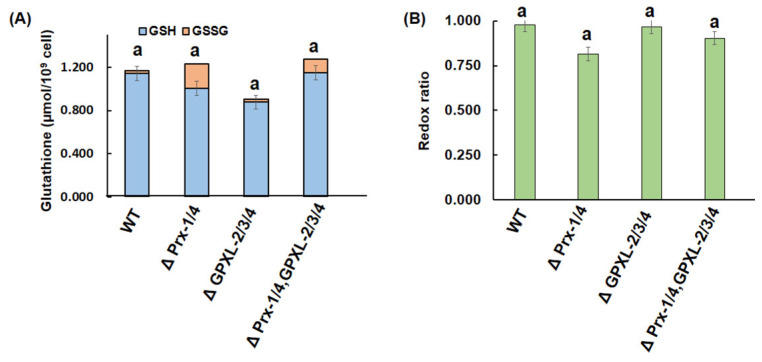
Quantitative measurement of glutathione (**A**) and its redox ratio (**B**) in mutants and wild-type cell lines of *Euglena*. ΔPrx-1/4, ΔGPXL-2/3/4, and ΔPrx-1/4, GPXL-2/3/4 corresponded to the cell lines where *Prx-1*/*Prx-4*, *GPXL-2*/*GPXL-3*/*GPXL-4*, and *Prx-1*/*Prx-4*, *GPXL-2*/*GPXL-3*/*GPXL-4* genes were suppressed, respectively. Total, reduced, and oxidized glutathione were measured, and no significant difference was observed in any cell lines compared to the wild type. Values with the same letter indicated no significant difference in glutathione content in mutant cell lines compared to the wild-type, according to Dunnet *t*-test (*p* < 0.05).

**Figure 12 biomolecules-14-00765-f012:**
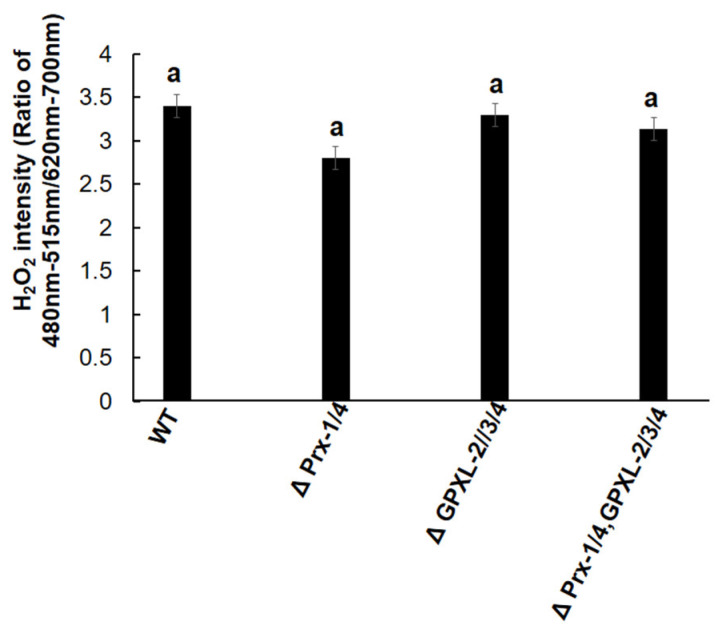
H_2_O_2_ quantitation in mutants and wild-type cell lines of *Euglena*. ΔPrx-1/4, ΔGPXL-2/3/4, and ΔPrx-1/4, GPXL-2/3/4 corresponded to the cell lines where *Prx-1*/*Prx-4*, *GPXL-2*/*GPXL-3*/*GPXL-4*, and *Prx-1*/*Prx-4*, *GPXL-2*/*GPXL-3*/*GPXL-4* genes were suppressed, respectively. Flurointensity among all the cell lines was determined by fluorescence detection system (Corona-SH 9000) using H_2_O_2-_specific fluorescent probe known as BES- H_2_O_2_-Ac (Derivative of BES (benzenesulfonyl) (Wako, Japan)). Values with same letter indicated no significant difference in H_2_O_2_ content in mutant cell line compared to wild type, according to Dunnet *t*-test (*p* < 0.05).

**Table 1 biomolecules-14-00765-t001:** Comparison of kinetic parameters among different peroxides obtained from different sources.

Organisms		Substrate	*V*max (µmol/min/mg Protein)	*K*m (µM)	*k*cat(s^−1^)	*k*cat/*K*m (M^−1^ s^−1^)	Reference
** *E. gracilis* **	rEgGPXL-2	H_2_O_2_	3.33 ± 1.0	71.43 ± 2.5	0.44 ± 0.33	6.1 × 10^3^	Present study
		*t*-BOOH	1.0 ± 0.2	28.5 ± 3.1	0.03 ± 0.01	1.0 × 10^3^	
	rEgGPXL-4	H_2_O_2_	10 ± 0.9	25 ± 0.4	0.33 ± 0.03	1.3 × 10^4^	
		*t*-BOOH	5.0 ± 1.1	38.5 ± 1.6	0.17 ± 0.03	4.4 × 10^3^	
	rEgGPXL-1	H_2_O_2_	3.03 ± 0.74	7.14 ± 2.3	0.505 ± 0.21	7.0 × 10^4^	[27]
		*t*-BOOH	4.25 ± 0.95	8.0 ± 2.7	0.708 ± 0.15	8.9 × 10^4^	
	rEgPrx-1	H_2_O_2_	3.15 ± 0.34	39.1 ± 4.1	1.21 ± 0.13	3.1 × 10^4^	[24]
		*t*-BOOH	2.21 ± 0.21	24.5 ± 4.9	0.85 ± 0.08	3.5 × 10^4^	
	rEgPrx-2	H_2_O_2_	3.34 ± 0.51	44.7 ± 3.1	1.28 ± 0.19	2.9 × 10^4^	
		*t*-BOOH	2.92 ± 0.13	38.8 ± 1.8	1.12 ± 0.05	2.9 × 10^4^	
		H_2_O_2_	2.29 ± 0.16	37.8 ± 6.5	0.88 ± 0.06	2.3 × 10^4^	
		*t*-BOOH	2.22 ± 0.11	36.1 ± 2.6	0.85 ± 0.04	2.4 × 10^4^	
	rEgPrx-3	H_2_O_2_	0.59 ± 0.04	3.4 ± 0.5	0.22 ± 0.01	6.6 × 10^4^	
		*t*-BOOH	0.77 ± 0.09	12.1 ± 1.4	0.29 ± 0.04	2.4 × 10^4^	
	rEgPrx-4	H_2_O_2_	3.34 ± 0.51	44.7 ± 3.1	0.44 ± 0.33	6.1 × 10^3^	
		*t*-BOOH	2.92 ± 0.13	38.8 ± 1.8	0.03 ± 0.01	1.0 × 10^3^	
	* GPX	H_2_O_2_	-	300	-	-	[47]
		*t*-BOOH	-	500	-	-	
** *C. reinhardtii* **	CrGPX-3	H_2_O_2_	0.26 ± 0.01	-	7.4 ± 0.2	137 × 10^3^	[11,51]
		*t*-BOOH	0.27 ± 0.06	-	11.3 ± 0.7	16 × 10^3^	
	CrGPX-5	H_2_O_2_	0.31 ± 0.06	54 ± 5	-	-	
		*t*-BOOH	0.32 ± 0.02	732 ± 112	-	-	
** *A. thaliana* **	AtGPX-8	H_2_O_2_	0.39 ± 0.02	65 ± 5.4	-	-	[17]
		*t*-BOOH	0.30 ± 0.01	-	-	-	

Note: All peroxidases use Trx as a reducing agent, except tetrameric GPX from *Euglena* (using GSH as a reducing agent marked with *). Values are the mean ± SD (*n* = 3).

## Data Availability

Sequencing data are available at the DDBJ Sequence Read Archive (DRA) with accession numbers SRP060591 and GDJR00000000.1 (GenBank). The comp numbers for EgGPXL-1, -2, -3, and -4 were comp13015, comp16152, comp26099, and comp39029, respectively. Other data related to this article will be shared upon reasonable request from the corresponding author.

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
