# Peer review of "Biochemical and Functional Profiling of Thioredoxin-Dependent Cytosolic GPX-like Proteins in Euglena gracilis"

_biomolecules, 2024, doi:10.3390/biom14070765_

Round 1
Reviewer 1 Report
Comments and Suggestions for Authors
In this article the authors able to fully characterize full-length cDNA obtained from the three cytosolic EgGPXLs via various biochemical and functional methods. Further, they showed that the these EgGPXLs used thioredoxin instead of glutathione as electron donor for reduction of both H2O2 and t-BOOH. The specific peroxidase activity for H2O2 and t-BOOH of these enzymes were 1.3 to 4.9 and 0.79 to 3.5 µmol/min/mg protein respectively. The research reported here is well characterized and can be accepted after the following minor revisions:
1. In the Figure 3 and 4, author needs to add error bar in the graph with triplicate experiments.
2. The introduction sections need to be little more elaborated.
Author Response
We are sincerely thankful to you for the careful analysis of our manuscript and the suggestions for further improvement. Our response is as follows. All the changes made in the revised manuscript are indicated by red fonts.
- In the Figure 3 and 4, author needs to add error bar in the graph with triplicate experiments.
Response: Error bar has been added in Figure 3 and 4 with triplicate experiments
- The introduction sections need to be little more elaborated.
Response: Some information has been added in introduction.
Reviewer 2 Report
Comments and Suggestions for Authors
The novelty and the quality of the manuscript are good and it does not need extensive improvement before publication. It is carefully organized and written. It is easy to follow it and contains clear comments and conclusions. In my opinion, this manuscript is very detailed and meticulous, it covers all the literature in the field with critical point of view. The topic have been completely covered and is well connected through the text. There is a significant novelty in presented topic. For all these reasons, I can recommend the acception of the manuscript after minor revision:
The manuscript should be extended in scientific discussion. The authors presented their results and compared to some works, but did not present explanations for the reasons to reach these results.
Author Response
We are sincerely thankful to you for the careful analysis of our manuscript and the suggestions for further improvement. Our response is as follows. All the changes made in the revised manuscript are indicated by red fonts.
- The manuscript should be extended in scientific discussion. The authors presented their results and compared to some works, but did not present explanations for the reasons to reach these results.
Response: We value your critical feedback regarding the requirement of more explanation of our results. After review carefully the whole manuscript, we have added some more information in section 3.1, 3.2, 3.3 and 3.4. Moreover, we believe that we have already provided detailed explanations for each result throughout the manuscript. We hope you are satisfied with our revision.